# TLR4 competence and mouse models of sublethal leptospirosis

Olifan Zewdie Abil ☺, Suman Kundu☺, Leonardo Moura Midon, Maria Gomes-Solecki *

Department of Microbiology, Immunology and Biochemistry, University of Tennessee Health Science Center, Memphis, United States of America,

☺ Equal contributions
* mgomesso@uthsc.edu

## Abstract

Mice are slowly being accepted as alternative models for investigation of leptospiral infection. The strain often used to analyze sublethal disease (C3H/HeJ) expresses a hyporesponsive *tlr4* gene in its cells and thus the model is deemed immunocompromised. To help resolve this scientific concern we compared infection of mice expressing competent *tlr4* (C3H/HeN, C57BL6) versus *tlr4* hyporesponsive mice (C3H/HeJ) with *Leptospira interrogans* serovar Copenhageni strain Fiocruz L1-130 over a period of two weeks. We found that the two mouse strains with a functional *tlr4* gene (C3H/HeN and C57BL/6) developed clinical and molecular signs of leptospirosis less pronounced but not significantly different than *tlr4* hyporesponsive C3H/HeJ, as quantified by weight loss, survival curves, presence of *Leptospira* 16S rRNA in blood and urine and burden of viable spirochetes in kidney as compared to the respective uninfected controls. Analysis of serologic immune factors in the three strains revealed increased IgM and IgG3, and a general absence of inflammatory markers at two weeks post infection. Our data suggests that TLR4 function is not sufficient to cause susceptibility to leptospirosis. We conclude that C3H/HeN and C57BL/6 are appropriate mouse models of sublethal leptospirosis.

## Author summary

We did a comparative study using mouse strains immunocompetent and hyporesponsive to *tlr4*. The data shows that *tlr4* competent strains (C3H/HeN and C57BL/6) developed clinical and molecular signs of sublethal leptospirosis not much different than *tlr4* hyporesponsive C3H/HeJ. Thus, competent recognition of *L. interrogans* serovar Copenhageni FioCruz factors by murine TLR4 does not determine susceptibility to leptospirosis.

**Data availability statement:** A Source data excel file is submitted as Supporting Information.

**Funding:** This work was supported by National Institute of Allergy and Infectious Diseases grant numbers R21 AI142129 (to MGS), R44 AI167605 (to MGS) and R01 AI175417 (to MGS). The funders had no role in study design, data collection and analysis, decision to publish, or preparation of the manuscript. OZA, SK, LMM and MGS received salary from the funders.

**Competing interests:** The authors have declared that no competing interests exist.

## Introduction

Leptospirosis, an often overlooked but resurging infectious disease caused by a spirochete, is a significant global health concern impacting millions of individuals worldwide. This disease carries a high mortality rate, resulting in approximately 65,000 deaths annually [1]. Furthermore, it poses a serious threat to animals of agricultural importance, leading to substantial economic losses, particularly in tropical and subtropical regions [2]. Although significant efforts have been made to formulate novel vaccination approaches that confer enduring immunity and safeguard against various serovars, our understanding of the specific immune factors contributing to host defense and disease progression remains limited [3].

Leptospirosis research is challenging due to the inconsistent outcomes observed in leptospiral infections involving animals and humans [2], and due to an abundance of high and low virulence *Leptospira* serovars [4–7]. Hamsters and guinea pigs have been widely used as small-animal models of acute disease, as they recapitulate the manifestations of severe disease in humans [8,9]. Rats have been used as a model for studying the severity of human leptospirosis as they become chronically infected and shed *Leptospira* in urine many months after infection [10]. However, the usage of these animals in research is limited in some parts of the world where the disease is endemic due to stringent animal regulations [11] and the scarcity of accessible reagents for routine experiments.

The outcomes of experimental leptospiral infection have been analyzed using various mouse models, including studies on lethal, sublethal, and chronic leptospirosis [12–14]. For instance, C57BL/6 and BALB/c are more resilient to acute disease and have the potential to serve as models for persistent infection caused by *Leptospira interrogans* [13,14]. C3H/HeJ mice infected with *L. interrogans* develop disease that can be easily monitored through measurement of clinical scores. This strain produces valuable models of lethal [12,15,16] and sublethal disease [17,18], and has been used to study inflammatory signatures of infection [19], necroptosis [20], and immunity protection [21].Toll-like receptor (TLR) 4, which recognizes LPS, plays a central role in the control of leptospirosis [22,23]. One study described how leptospiral-LPS activates murine, but not human, TLR4 in cultured macrophages [24], and it is associated with resistance to infection [25]. C3H/HeJ mice have a single amino acid substitution (aa712, P to H) within the coding region of the *tlr4* gene that makes this molecule hyporesponsive to the atypical *Leptospira* LPS [23,26]. Of note, humans, also believed to be susceptible to leptospirosis express a TLR4 molecule that does not sense the atypical *Leptospira* LPS [24]. Nevertheless, one consistent criticism regarding the use C3H/HeJ mice is that its hyporesponsive Toll-Like Receptor 4 (TLR4) [26] qualifies these mice as immunocompromised. To help resolve this valid scientific concern we did a study in which we compared infection with *L. interrogans* in mice expressing competent *tlr4* (C57BL6, C3H/HeN) versus mice that are *tlr4* hyporesponsive (C3H/HeJ). The goal of this study was to determine if the TLR4 competent strains can also be used as mouse models of sublethal leptospirosis.

## Results

### Weight loss, burden in blood, shedding in urine and survival after infection with *L. interrogans* serovar Copenhageni FioCruz.

Following experimental leptospiral infection ($10^8$ LiC), mice were monitored over a 15-day period. In C3H/HeJ, steady weight loss was observed beginning on day 6, reaching the lowest mean on day 9 (-14.9%), at which point mice recovered gradually (Fig 1A). C3H/HeN mice exhibited weight loss from day 1 reaching the lowest mean on day 7 (-11.9%) which was maintained until day 14 (Fig 1B). C57BL/6 mice experienced lower loss of weight throughout the 14 days, reaching the lowest mean (-7.4%) on day 6 (Fig 1C). The weight loss curves of the three infected mouse strains were statistically significant compared to their respective controls ($p < 0.0001$).

Quantitative PCR (qPCR) targeting the *Leptospira* 16S rRNA gene was performed to analyse the dissemination of *Leptospira* in blood and urine. Blood samples were collected on days 0, 1, 3, 6, 9, 12, and 14 post-inoculation to quantify the *Leptospira* burden. Peak *Leptospira* burdens in blood were observed on day 3 in C3H/HeJ mice ($\sim 3 \times 10^4$), whereas the peak burden occurred on day 1 in both C3H/HeN and C57BL/6 mice, $\sim 6$-$9 \times 10^3$ (Fig 1D). *Leptospira* shedding in urine was also assessed using qPCR on alternate days. Shedding (Fig 1E) was detected at low levels ($\sim 10^2$) during the first week post-infection and gradually increased in the second week with peaks on day 10 for C3H/HeN and C57BL/6 ($\sim 10^4$), and on day 14 for C3H/HeJ ($\sim 10^6$).

Regarding survival, none of the 9–10 week old mouse strains injected with $10^8$LiC reached the LD50: 57.1% of C3H/HeJ, 71.4% of C3H/HeN and 85.7% C57BL/6 survived at 14 days post-infection in comparison with 100% survival in each of the control groups (Fig 2). Differences are not statistically significant.

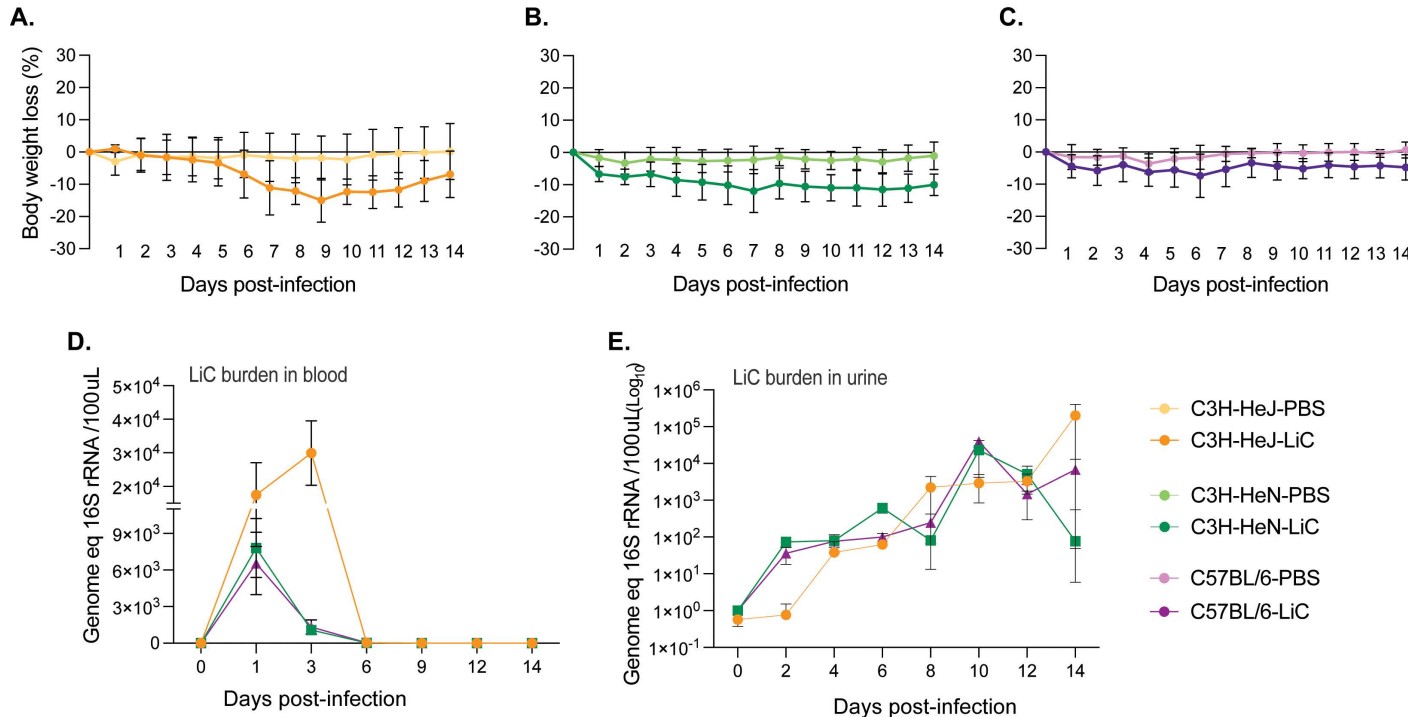

**Fig 1. Weight loss and bacterial burden in blood and urine after infection.** Male C3H/HeJ, C3H/HeN and C57BL/6 mice (n = 7/group) were inoculated IP with $10^8$ *L. interrogans* serovar Copenhageni strain FioCruz L1-130 (LiC) and with PBS as control. A-C) Body weight (% change) was recorded daily for 15 days post infection; D-E) Quantification of bacterial load in blood and urine by qPCR of the 16S rRNA gene. Data from two independent experiments is shown. For weight loss, statistics by Ordinary 2-Way ANOVA, p < 0.0001 for infected versus control C3H/HeJ, C3H/HeN and C57BL/6.

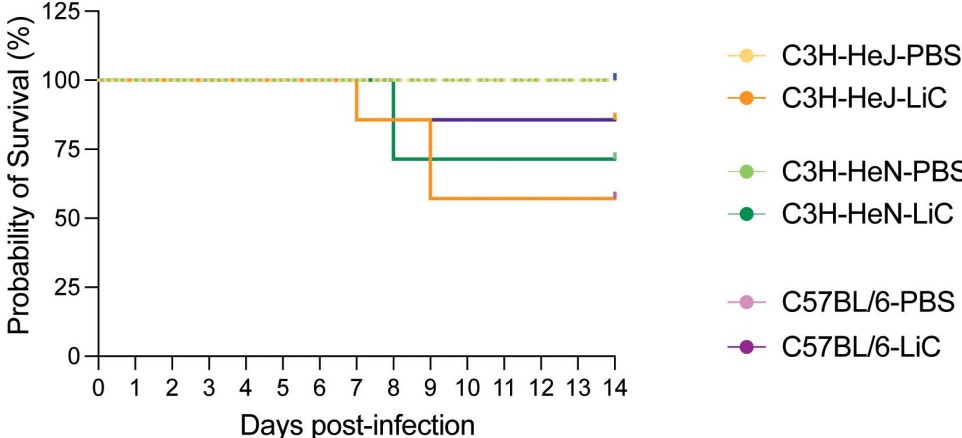

**Fig 2. Probability of survival.** Male C3H/HeJ, C3H/HeN and C57BL/6 mice (n = 7/group) were inoculated IP with $10^8$ *L. interrogans* serovar Copenhageni strain FioCruz L1-130 (LiC) and with PBS as control. The percentage of mice that met endpoint criteria for euthanasia before and at the term of two independent experiments is shown. Statistics by Log-rank (Mantel-Cox) test, p = 0.1088.

### Burden and viability of *L. interrogans* serovar Copenhageni FioCruz in kidney tissue

To assess renal colonization by *L. interrogans*, kidney tissues from infected mice were collected on day 15 post-infection and analyzed for the presence of live *Leptospira* under a dark field microscope and for DNA quantification using qPCR. The bacterial burden was ~ $7.3 \times 10^4$ *Leptospira* per mg of kidney tissue in C3H/HeJ, ~ $4.1 \times 10^4$ in C3H/HeN, and ~$5.4 \times 10^4$ in C57BL/6 infected mice (Fig 3A). Differences between infected and control groups are significant, p < 0.001. Additionally, the viability of *Leptospira* was assessed by culturing kidney tissues in EMJH medium at 30°C for 4 days, visualizing live motile *Leptospira* under a dark field microscope and quantifying spirochetes using qPCR. On day 4 of culture, the average number of spirochetes per 100 µL of EMJH culture was~ $4.3 \times 10^3$ for C3H/HeJ, ~ $3.1 \times 10^3$ for C3H/HeN, and ~$2.2 \times 10^3$ for C57BL/6 mice. Differences between infected and control groups are significant, p < 0.05 (Fig 3B).

### Inflammatory cytokines and signatures of fibrosis in kidney

To assess the upregulation of inflammatory cytokines critical for immune responses and fibrosis markers associated with kidney damage, kidney was collected from experimental mice on day 15 post-infection and processed for mRNA expression analysis of TNF-α, TGF-β1, IL-1β, IFN-γ, IL-6, IL-23, IL-17a, iNOS and ColA1 by RT-PCR using glyceraldehyde 3-phosphate dehydrogenase (GAPDH) as the calibrator to normalize gene expression. Expression of TNF-α, IL-1β, IFN-γ, IL-6, IL-23, IL-17a were not different between infected and uninfected groups. The expression TGF-β1 mRNA, was upregulated in C3H/HeJ mice in comparison to uninfected control; however differences between the mouse strains were not significant (Fig 4A). In addition, mRNA of the fibrosis marker inducible Nitric Oxide Synthase (iNOS), was increased in C3H/HeJ, C3H/HeN and C57BL/6 in comparison to the respective controls; however differences between the mouse strains were not significant (Fig 4B). The mRNA expression of collagen A1 (ColA1) was elevated in both C3H/HeJ and C3H/HeN mice compared to the respective controls in contrast to C57BL/6, and differences between the mouse strains were significant (Fig 4C).

### Inflammatory markers circulating in serum

The concentration of chemokines CxCL1/KC/GRO-α, CxCL2/MIP-2, CCL5/RANTES and cytokines TNF-α, IL-1β, IFN-γ, IL-5, IL-6, IL-9, IL-10, IL-13, IL-17, IL-23 circulating in blood were measured in experimental animals on day 15 post-infection. Only one chemokine, CCL5/RANTES, was increased in infected C3H/HeJ in comparison to the respective

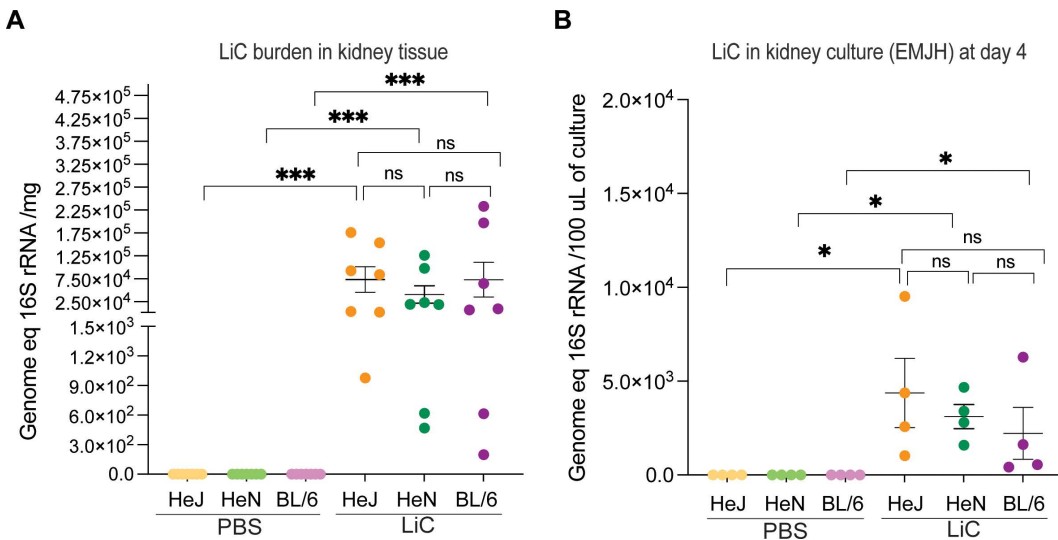

**Fig 3. Burden and viability of *Leptospira* in kidney.** A) Kidney tissues were collected 2 weeks post-infection for qPCR analysis of the 16S rRNA gene; data from two independent experiments is shown; B) kidney was placed in culture to evaluate motility of *L. interrogans* under a dark field microscope on day 4, and qPCR was used to quantify *Leptospira*; data from one experiment is shown. Statistics by Mann-Whitney U test, *** p < 0.001 and * p < 0.05.

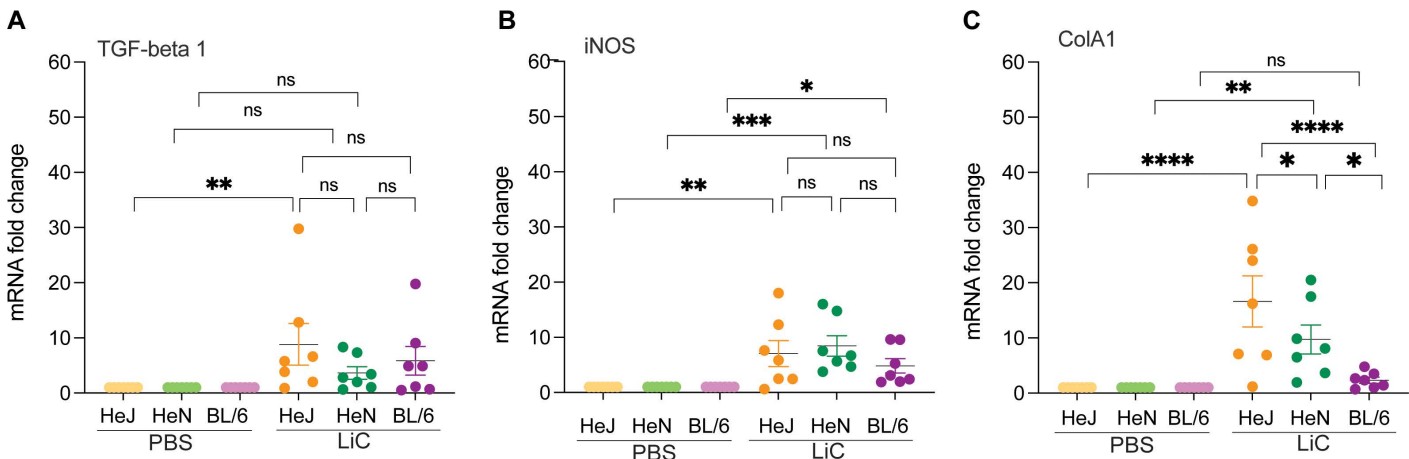

**Fig 4. Inflammatory and fibrosis markers in kidney.** Kidney tissues collected 2 weeks post-infection were evaluated for mRNA expression analysis of 9 inflammatory markers by RT-PCR normalized against GAPDH. We show TGF-beta 1 (A), iNOS (B) and ColA1 (C) as these markers produced differences between the three mouse strains in two independent experiments. Statistics by Two-way ANOVA followed by Tukey's multiple comparisons test, * p < 0.05, ** p < 0.005, *** p < 0.0005, **** p < 0.0001.

uninfected control. None of the inflammatory markers tested were increased in serum from C3H/HeN or C57BL/6 two weeks post infection. Differences in CCL5/RANTES between C3H/HeJ and C3H/HeN, as well as C3H/HeJ and C57BL/6 were significant (Fig 5).

## Serological antibody responses

Terminal blood samples were collected 15 days post-infection to evaluate the levels of antibody production to *L. interrogans* (Fig 6). As expected, the three strains of infected mice produced increased levels of anti-*Leptospira*-specific IgM

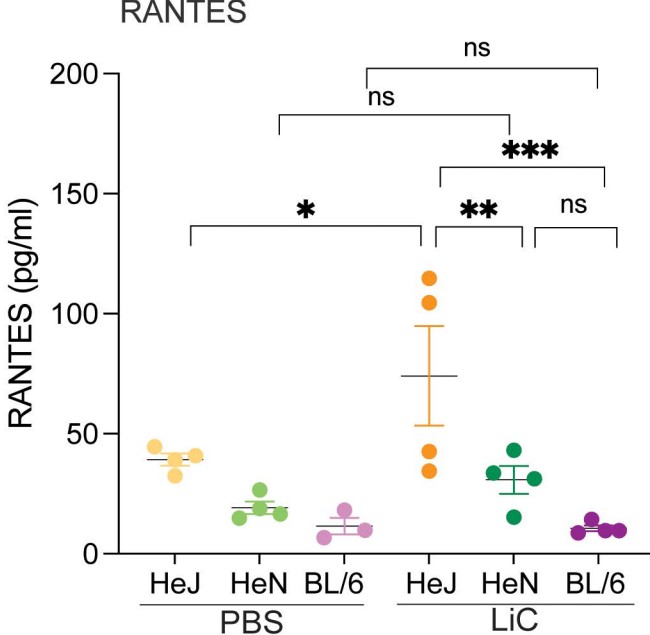

**Fig 5. Inflammatory markers circulating in serum.** The concentration of 13 inflammatory markers was measured in serum from experimental mice collected 2 weeks post-infection. Shown is RANTES, the only chemokine detected. Data from one experiment is shown. Statistics by Two-way ANOVA followed by Tukey's multiple comparisons test, *p<0.05, ** p<0.005, ***p<0.0005.

(Fig 6A) and total IgG (Fig 6B) compared to the respective uninfected controls; no differences were observed between the mouse strains. IgG isotyping produced the following results on d15 post-infection: IgG3 mirrored the total IgG profile for the three mouse strains (Fig 6C) and IgG1/IgG2a were not detected in infected C57BL/6 compared to the uninfected control (Fig 6D and 6E). IgG1 was increased in infected C3H/HeJ and C3H/HeN compared to the respective controls but differences between the C3H mouse strains were not significant (Fig 6D). IgG2a was only increased in infected C3H/HeN compared to the uninfected control, and differences between this strain and C3H/HeJ as well as C57BL/6 were significant (Fig 6E).

## Discussion

We previously used the C3H/HeJ strain to develop a mouse model of experimental sublethal leptospirosis [17,27]. Since resistance to acute lethal infection with *L. interrogans* was associated with a functional *tlr4* [25] and C3H/HeJ is hyporesponsive [26] to *L. interrogans* atypical LPS, there are valid concerns that this mouse model does not recapitulate a competent immune response to this spirochete. In this study, we did a side-by-side comparative analysis using three mouse strains (C3H/HeJ, C3H/HeN and C57BL/6). The two mouse strains with a functional *tlr4* gene (C3H/HeN and C57BL/6) developed clinical and molecular signs of sublethal leptospirosis that were less pronounced but not significantly different than C3H/HeJ. Our data indicates that TLR4-competent C3H/HeN and C57BL/6 are appropriate mouse models of sublethal leptospirosis.

We found that weight loss differences between strains are more evident in C3H mice throughout the 15-day infection period (**Fig 1**). For a choice of animal models of disease, differences in weight loss in the C3H background produced clinical scores that may be easier to reproduce consistently. qPCR analysis of *Leptospira* 16S rRNA in blood, urine and kidney tissue showed no differences in the dynamics of *L. interrogans* dissemination between the three mouse strains, except for

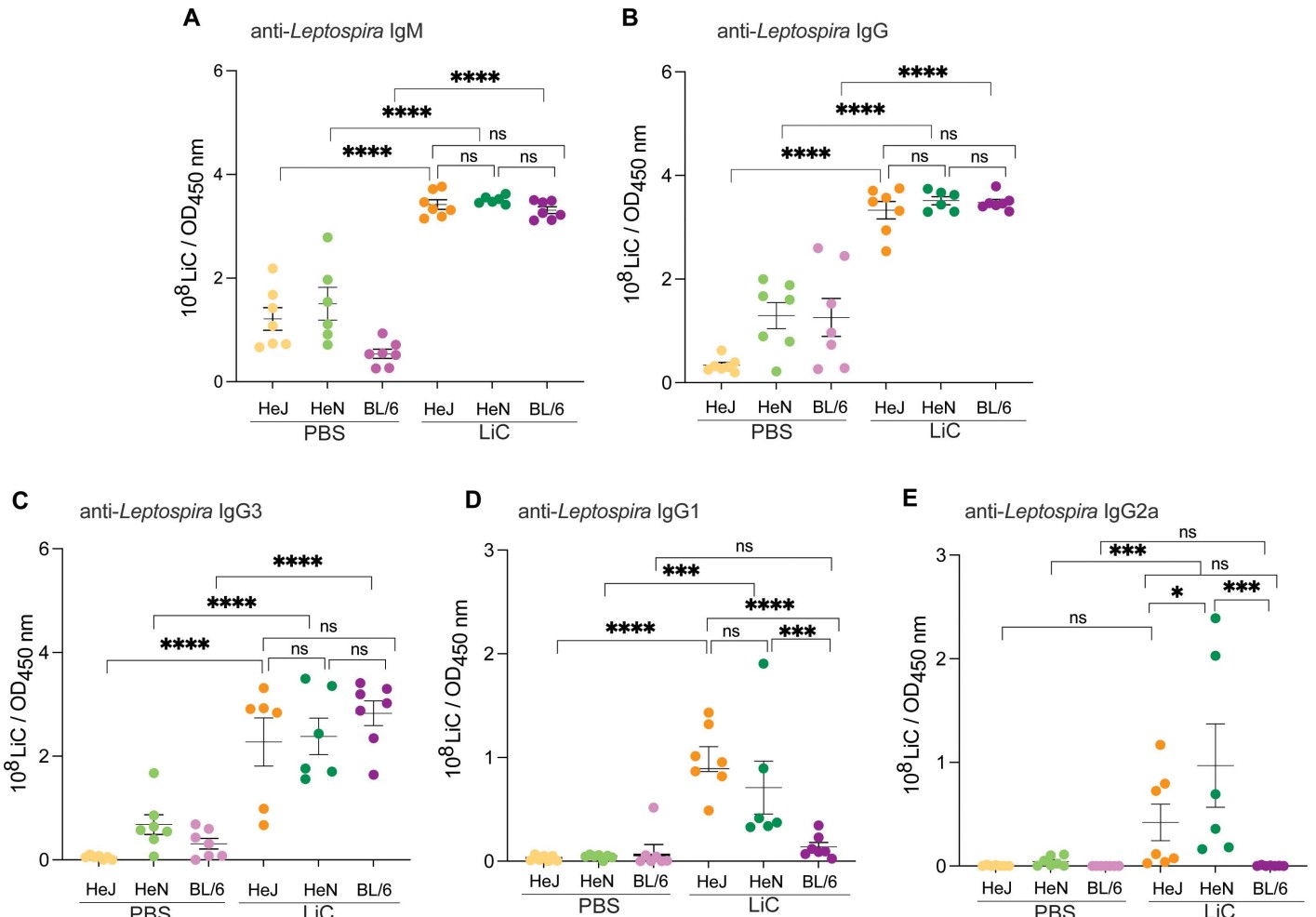

**Fig 6. Antibody responses to *Leptospira*. *L. interrogans* serovar Copenhageni FioCruz specific IgM, IgG, IgG3, IgG1 and IgG2a were measured in serum from experimental mice by ELISA on day 15 post-infection.** Data from two independent experiments is shown. Statistics by Two-way ANOVA with uncorrected Fisher's LSD test, * p < 0.05, *** p < 0.0005, **** p < 0.0001.

the load of *Leptospira* in blood which was higher in 9–10 week old C3H/HeJ (**Figs 1** and **3**). These data reproduce previous studies done using age/IP-dose matched C3H/HeJ [18] and C57BL/6 [28] mice.

There are numerous links between competent recognition of Leptospiral LPS by TLR4 and resistance to infection in mice [13,22–25]. Our data shows that susceptibility to sublethal infection can be observed in 9–10 week old C3H/HeN and C57BL/6 mice, which express competent TLR4 (**Figs 1** and **2**). Nevertheless, our data also shows that C3H-HeJ, expressing a hyporesponsive *tlr4*, consistently produced larger metrics of disease progression. Of note, humans express a *tlr4* molecule that does not sense the atypical *Leptospira* LPS [24]. One could therefore speculate that the C3H/HeJ mouse may recapitulate the human immune response to pathogenic *Leptospira* more closely than *tlr4* competent mice. In another study we found that inoculation of potentially susceptible TLR4/MD-2 humanized transgenic mice with *L. interrogans* did not produce different measurements of disease progression or *Leptospira* dissemination than wildtype mice [28]. This may have been related to the humanized transgenic mouse background being C57BL/6. Overall, the data suggests

that although TLR4 plays a role in susceptibility to *Leptospira* infection, it is not the main determinant as other receptors in mammalian cells (ex. TLR2) likely compensate for deficient receptor binding.

We analysed mRNA expression of 9 inflammatory markers (TGF-β1, TNF-α, IL-1β, IFN-γ, IL-6, IL-23, IL-17a, iNOS and ColA1) in kidney of experimental mice at 15 days post infection (**Fig 4**). We found that C3H/HeJ had increased TGF-beta 1, iNOS and ColA1, C3H/HeN had increased iNOS and ColA1, and C57BL/6 has iNOS increased but not ColA1. When we compare the C3H/HeJ data with a previous study using age/IP-dose matched mice studies [18], normalized for markers tested in both studies, we find the iNOS and ColA1 data in agreement, but not TNF-α and IFN-γ. Similarly, when we compare the C57BL/6 data with a previous study using age/IP-dose matched mice [28] we find discrepancies in iNOS, ColA1 and TNF-α. These differences suggest that although expression markers of inflammation in kidney may be statistically different in the reported studies, it still raises a question of biological relevance and should be interpreted with caution.

Regarding inflammatory markers circulating in serum we tested for markers engaged in innate (CxCL1/KC/GRO-α, CxCL2/MIP-2, CCL5/RANTES, TNF-α, IL-1β) and adaptive Th1, Th2 and Th17 responses (IFN-γ, IL-5, IL-6, IL-9, IL-10, IL-13, IL-17, IL-23) and found that only CCL5/RANTES was increased in C3H/HeJ at day 15 post infection (**Fig 5**). CCL5/RANTES is a chemokine important for the recruitment of T cells, neutrophils and NK cells. In our previous studies using age/IP-dose matched C3H/HeJ, we found that CCL5/RANTES was increased in serum at 24h and 72h post sublethal infection [19]. Furthermore, CCL5/RANTES was found increased in kidney, spleen and blood of C57BL/6J mice at 72h post-infection with 108 *L. interrogans* serovar Manilae in a model of lethal leptospirosis [29]. Now looking at the human leptospirosis landscape on CCL5/RANTES, low levels of serologic CCL5/RANTES were found in patients with fatal leptospirosis [30] and CCL5/RANTES was the most increased chemokine found in serum of patients diagnosed with non-lethal leptospirosis associated with *L. interrogans* serovars Copenhageni and Icteroheamorrhagiae [31]. Overall, the data from *Mus* and *Homo* strongly suggests that CCL5/RANTES plays an important role in the immune response to *L. interrogans* infection. We will further characterize this chemokine in future analysis of inflammatory markers of lethal and sublethal leptospirosis.

While innate immune responses are the main immunological pathway for eliminating *Leptospira*, the humoral immune response plays a vital role in effectively eradicating the bacteria and expelling it from the host [32]. Our present observations (**Fig 6**) revealed that LiC infection led to an increase of *Leptospira*-specific IgM, IgG antibodies in the blood of the three mouse strains, which was expected [17,25,33,34]. IgG isotyping produced some interesting results for IgG3 and IgG1. IgG3, which is T-cell independent, was significantly increased in the three mouse strains and explains high total IgG for C57BL/6 in our study. Two weeks after infection, IgG1 was extremely low in C57BL/6, as observed by others [34], and significantly higher in C3H/HeJ and C3H/HeN. The production of IgG1 antibody in C3H mice suggests a Th2-biased engagement of the immune response at two weeks post infection, as shown in our previous studies in C3H/HeJ [17].

In conclusion, our data shows that C3H/HeN and C57BL/6 mice, both TLR4 competent strains, can be used to recapitulate sublethal leptospirosis as they produce unambiguous differences in clinical and molecular measurements of disease progression, *Leptospira* dissemination to tissues, colonization of kidney by live spirochetes and shedding in urine. Thus, competent recognition of *L. interrogans* serovar Copenhageni FioCruz factors by murine TLR4 does not determine susceptibility to disease. Each mouse strain has characteristics that can be leveraged in pursuit of knowledge on immunity and host response to pathogenic *Leptospira* species.

## Materials and methods

### Ethics Statement

All experiments with animals were performed in compliance with the University of Tennessee Health Science Center (UTHSC) Institutional Animal Care and Use Committee (IACUC), Protocol no. 22–0362.

## Animals

Age matched 9–10 weeks, male C3H/HeJ, C3H/HeN, C57BL/6 mice were used. C3H/HeJ animals were purchased from The Jackson Laboratory (Bar Harbor); C3H/HeN and C57BL/6 were purchased from Charles River (Wilmington, MA) and maintained in specific pathogen-free at the Laboratory Animal Care Unit of the University of Tennessee Health Science Centre, with unrestricted access to food and water. Male mice were used because they are more susceptible to leptospirosis. Two independent experiments were done: the first was comprised of 6 groups of mice with 3 animals per group; the second, was comprised of 6 groups of mice with 4 animals per group; data from both experiments were combined for analysis.

## Bacterial strains and culture

*L. interrogans* serovar Copenhageni strain Fiocruz L1-130 (henceforth LiC), frozen in -80°C, was passaged in hamster. Their kidneys were harvested and cultivated in 4mL of Hornsby-Alt-Nally (HAN) media [35] with 100 µg/mL 5-fluorouracil (MP Chemicals, CA) at 29°C for better growth. Passage 2 was done in Ellinghausen-McCullough-Johnson-Harris (EMJH) medium supplemented with Difco *Leptospira* enrichment EMJH (Becton, MD) at 28–30°C. EMJH culture passage 2 was allowed to reach the log phase of growth, pelleted by centrifugation at $3,000 \times g$ for 5 min, and washed and resuspended in sterile $1 \times$ phosphate-buffered saline (PBS) (Thermo Fisher Scientific). Next, the cells were used to infect the animals after counted under a dark-field microscope (Zeiss USA, Hawthorne, NY) using a Petroff- Hausser chamber as previously described [27]. We found that recovering -80C frozen *Leptospira* in HAN media before passage in EMJH allowed for faster recovery of the culture.

## Animal infection and collection of specimens

Mice were inoculated with $1 \times 10^8$ LiC in 200 µL of endotoxin-free $1 \times$ PBS intraperitoneally (i.p.). Control mice were inoculated with the same volume of endotoxin-free $1 \times$ PBS. Survival and body weight loss were monitored for 15 consecutive days post-infection. Mice were euthanized 15 days post-infection or when they reached the humane endpoint criteria (20% body weight loss). Blood, urine, and kidney were collected for further analysis. Immediately after the mice were restrained, the bladder was massaged, and Eppendorf tubes were used to collect the excreted urine.

## Bacterial quantification

The NucleoSpin tissue kit (Clontech, Mountain View, CA) was used to purify genomic DNA from kidney, blood and urine following the manufacturer's protocol, and the purified DNA was then stored at −20°C for further analysis. To quantify *Leptospira*, quantitative polymerase chain reaction (qPCR) was performed using *Leptospira* 16S rRNA primers (Forward: CCCGCGTCCGATTAG and Reverse: TCCATTGTGGCCGAACAC) and a TAMRA probe (CTCACCAAGGCGACGATCG-GTAGC) obtained from Eurofins (Huntsville, AL). The results were reported as the number of *Leptospira* genome equivalents. The qPCR mixture consisted of 25 µM of each primer, 250 nM of the specific probe, and 2 µL of DNA sample, with a total volume of 20 µL. Duplicate reactions were performed. The amplification protocol included an initial step of 10 min at 95°C, followed by 40 cycles of amplification (15 s at 95°C and 1 min at 60°C). The analysis was conducted using the comparative threshold cycle (CT) method. A negative result was determined if no amplification occurred.

## Measurement of *Leptospira*-specific antibody

*Leptospira*-specific antibody levels were measured using the enzyme-linked immunosorbent assay (ELISA). The process of preparing a leptospiral extract for LiC was followed as described [36]. Briefly, LiC was cultured in EMJH media until it reached optimal cell density. The bacterial cells were then separated by centrifugation to form a pellet. This pellet was subjected to incubation with BugBuster solution (1mL) at room temperature in a shaker (100 rpm) for 20 min and mixed thoroughly by

vortexing. The resulting mixture was stored at -20°C. The whole cell extract of *Leptospira* was appropriately diluted in a sodium carbonate coating buffer with a concentration of 1X. A 96-well flat-bottom ELISA microtiter plate (Nunc-eBioscience) was coated overnight at 4°C with 100 µL 1X sodium carbonate coating buffer whole-cell extract of *Leptospira* ($10^7$–$10^8$ bacteria per well). After overnight incubation, the ELISA plate was washed with 1X PBST. The plate was blocked by a blocking buffer (100 µL/well) containing 1% BSA, followed by incubation for 1 h at 37°C. After washing, serum samples (1:100) was added, and the plate was incubated for 1 h at 37°C. The unbound primary antibody was removed by vigorous washing. Next, anti-mouse secondary antibodies for IgM IgG, IgG3, IgG1 or IgG2a (all from Cell signaling technology, CST) conjugated with horseradish peroxidase was added, which was incubated for 30 min, followed by standard color development using TMB Sure-blue. Absorbance measurement was carried out at OD 450 nm using Molecular Devices Spetramax.

## Expression of inflammatory and fibrosis markers

Total RNA was extracted using the RNeasy Mini Kit (QIAGEN) from kidney according to the manufacturer's protocol. Complementary DNA (cDNA) was synthesized from the extracted RNA using the RevertAid First Strand cDNA Synthesis Kit (Thermo Fisher Scientific). The resulting first-strand cDNA served as the template for reverse transcription PCR (RT-qPCR), which was performed on a QuantStudio 3 Real-Time PCR (Applied Biosystems) using the PowerTrack SYBR Green Master Mix (Applied Biosystems). Each RT-qPCR reaction (10 µL total volume) included the cDNA template and specific primers. The cycling conditions were as follows: an initial step at 50°C for 2 minutes, followed by denaturation at 95°C for 2 minutes, followed by 40 cycles of 95°C for 15 sec for denaturation, and 60°C for 1 min for annealing/extension. A melt curve analysis was conducted at the end of the amplification to confirm the specificity of the PCR products. Relative gene expression levels across samples were quantified by the double delta Ct ($2^{-\Delta\Delta Ct}$) method, with glyceraldehyde 3-phosphate dehydrogenase (GAPDH) serving as the endogenous reference control. The primer sequences used are listed in S1 Table in the Source Data file.

## Measurement of chemokines and cytokines in blood

Serum from experimental mice was derived from blood collected after euthanasia on day 15 post-infection and frozen at -80C. Analysis of circulating chemokines and cytokines was done using ProcartaPlex Multiplex Immunoassay (eBioscience) according to the manufacturer's instructions. The data were acquired using a Luminex 200 reader. The concentrations of each inflammatory marker were determined based on a standard curve.

## Statistical analysis

For the exploratory animal research described in this study, we used the "resource equation" approach to calculate sample size. Based on this approach, the acceptable range of degrees of freedom (DF) for the error term in an analysis of variance (ANOVA) is between 10–20 animals [37]. This method allows for determination of the minimum number of animals to produce significant results. We used GraphPad Prism 10 software. For analysis of weight loss (Fig 1) we used Ordinary 2-Way ANOVA; for survival (Fig 2) we used Mantel-Cox Log rank test; for burden and viability of LiC (Fig 3) we used Mann-Whitney *U* test; for analysis of inflammatory markers (Figs 4, and 5) we used Two-way ANOVA with Tukey's multiple comparison test; for analysis of antibody class and IgG isotypes (Fig 6) we used Two-Way ANOVA with uncorrected Fisher's LSD.

## Supporting information

**S1 Table. Excel file containing the source data used in all figures, and S1 Table containing a list of primers for inflammatory and fibrosis markers.**
(XLSX)

## Author contributions

**Conceptualization:** Maria Gomes-Solecki.

**Data curation:** Olifan Zewdie Abil, Maria Gomes-Solecki.

**Formal analysis:** Olifan Zewdie Abil.

**Funding acquisition:** Maria Gomes-Solecki.

**Investigation:** Olifan Zewdie Abil, Suman Kundu, Leonardo Moura Midon.

**Methodology:** Olifan Zewdie Abil, Suman Kundu, Leonardo Moura Midon.

**Supervision:** Maria Gomes-Solecki.

**Writing – original draft:** Olifan Zewdie Abil, Suman Kundu, Maria Gomes-Solecki.

**Writing – review & editing:** Maria Gomes-Solecki.

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
