## [Decision Letter · Decision Letter 0]

5 Mar 2025

PNTD-D-25-00171

TLR4 competence and mouse models of leptospirosis

Dear Dr. Gomes-Solecki,

Thank you for submitting your manuscript to PLOS Neglected Tropical Diseases. After careful consideration, we feel that it has merit but does not fully meet PLOS Neglected Tropical Diseases's publication criteria as it currently stands. Therefore, we invite you to submit a revised version of the manuscript that addresses the points raised during the review process.

The Editors are very enthusiastic about this manuscript and hope for a suitably modified version soon. We do not ask for new experiments to be done. Please address these comments appropriately as potential limitations and that (as always) there is need for more experiments to be done.

Please submit your revised manuscript within 60 days May 04 2025 11:59PM. If you will need more time than this to complete your revisions, please reply to this message or contact the journal office at plosntds@plos.org. Please include the following items when submitting your revised manuscript:

We look forward to receiving your revised manuscript.

Kind regards,

Joe Vinetz

Section Editor

Shaden Kamhawi

co-Editor-in-Chief

Paul Brindley

co-Editor-in-Chief

**Journal Requirements:**

At this stage, the following Authors/Authors require contributions: Olifan Zewdie Abil, Suman Kundu, Leonardo Moura Midon, and Maria Gomes-Solecki. Please ensure that the full contributions of each author are acknowledged in the "Add/Edit/Remove Authors" section of our submission form.

- ® on page: 7.

4) Please ensure that the funders and grant numbers match between the Financial Disclosure field and the Funding Information tab in your submission form. Note that the funders must be provided in the same order in both places as well.

**Reviewers' Comments:**

Reviewer's Responses to Questions

**Key Review Criteria Required for Acceptance?**

**Methods:**

-Are the objectives of the study clearly articulated with a clear testable hypothesis stated?

-Is the study design appropriate to address the stated objectives?

-Is the population clearly described and appropriate for the hypothesis being tested?

-Is the sample size sufficient to ensure adequate power to address the hypothesis being tested?

-Were correct statistical analysis used to support conclusions?

-Are there concerns about ethical or regulatory requirements being met?

Reviewer #1: The methods and objectives are clearly stated with a testable hypothesis. The study is appropriate. Concerns are listed below.

However, two issues are noted:

1. The authors are making the claim that C3H/HeJ mice are the only strain that sustained lethal infection, however, no comparison was ever made using age-matched mice of the other strains. Therefore this statement cannot be made without proper comparisons. Experiments need to be done with 7-8 week mice of the other strains to allow direct comparison.

a. Line 31-32 – this experiment was done with the 7-8 week mice

b. Line 33-36 – This experiment was done with 9-10 week mice

c. The way lines 31-36 are written the authors state the C3H/HeJ mice had lethal infection, but then had similar survival curves - thus contradicting without stating these are different aged mice.

d. Apply this to other instances in the paper. Line 258-259, 276-277 etc.

2. Mouse experiments were only done with 4 mice and two replicates. A third replicate needs to be done for accurate and valid comparisons to be made, as this is the standard for scientific conclusions to be drawn.

Reviewer #2: The manuscript clearly states its objectives; however, the hypothesis could be more explicitly defined. The design is generally appropriate for addressing the stated objectives, but the use of n=4 mice per group is a limitation, potentially affecting statistical power and reproducibility. The sentence in lines 120-121, which states "We used 4 animals per group, and experiments were reproduced once" is ambiguous. The authors should clarify whether they conducted two independent experimental replicates or only a single experiment without replication, which would be a major limitation. Additionally, the reasoning behind using both HAN and EMJH media should be explained. The method of urine collection is not described and should be included in the methods section (lines 134-139). The age of the mice used for cytokine analysis is unclear; lines 230-232 read as if a different experiment was conducted for this, which should be clarified in the methods. Furthermore, duplicates are usually not sufficient for qPCR analysis, and it is unclear how absolute quantitation of lipL32 copies was calculated. The term "confluency" in line 157 needs further clarification. It is also unclear what tissue was used in the methods section for the expression of inflammatory and fibrosis markers. The reasoning for choosing specific inflammation and fibrosis markers is not presented, and the method used for their quantification is questionable. A major advantage of mouse models over hamsters and guinea pigs is the availability of assays such as flow cytometry and ELISAs for direct evaluation of circulating cytokines in mice. It is unclear why the study relied solely on RT-qPCR, a method also available for hamsters, despite its limitations. The statistical methods are appropriate but should be better justified in terms of their power to detect significant differences given the small sample size. The study adheres to ethical guidelines, but further clarification on humane endpoints and proper referencing of the literature would be beneficial to the reader.

**Results:**

-Does the analysis presented match the analysis plan?

-Are the results clearly and completely presented?

-Are the figures (Tables, Images) of sufficient quality for clarity?

Reviewer #1: 1. Figures: Ideally, all three replicates (when the final one is performed) should be averaged and presented as a combined figure. Otherwise you need to provide a strong reason why you aren’t combining them, and include the data from all three in the supplement. This will allow reproducibility to be seen and verified.

2. Line 214 – an explanation of why younger mice were all of a sudden chosen for analysis is needed, and then as above, experiments need to be completed with the other strains with same ages of mice.

3. Figures: Low quality images, when looking at it there seems to be pixelation causing additional faint lines to show up on graphs. Please provide better quality images.

4. Supplementary Figure 1: This would perhaps be best incorporated into Figure 1 so we can see the differences between the two ages of mice easily. This is an important conclusion you are drawing and it is in the supplement.

5. Figure 3B: qPCR of a culture following a harvest does not provide quantitative information on the burden in tissue. Cultures do not grow the same and therefore this is not a quantitative measure and should not be included. You can include a statement that cultures were positive, which is what matters.

Reviewer #2: The results are presented in a logical order, but some aspects need clarification. The 75% survival rate mentioned in results (line 211) and in Figure 2 is unclear. If two experiments were performed, as suggested by the legend for Figure 2, but only one mouse died, then the percentage calculation is incorrect. To prevent this kind of misunderstanding, results for the presumed second experiment should be displayed. In the paragraph in lines 203-213, the numbers in parentheses should be followed by an explanation of what they represent (leptospires?). Figures also need editing to improve data display. The simple timelines in Figures 2, 3, 4, and 5 are not necessary. The significance bars in Figure 3 should be adjusted to reduce the Y-axis, focusing on the data rather than dedicating excessive space to the significance bars. A similar issue is seen in Figures 4 and 5. Weight loss data in Figure 1 could be combined, though this is a minor concern.

**Conclusions:**

-Are the conclusions supported by the data presented?

-Are the limitations of analysis clearly described?

-Do the authors discuss how these data can be helpful to advance our understanding of the topic under study?

-Is public health relevance addressed?

Reviewer #1: Generally, conclusions are supported. However, as described above there is no evidence that 7-8 week old mice of the C3H/HeN or C57BL/6 mice would not result in a lethal infection. This needs to be addressed experimentally in order for the conclusion that the C3H/HeJ is the only lethal model of infection (and the caveat added on age of mice).

In the discussion, a section needs to be added discussing the importance of the RT-qPCR results. These were only referenced in the results section with no discussion of their importance or differences between strains and what impact that would have on the model. Please add this to the manuscript.

Reviewer #2: The major limitations of this study are sample size and reproducibility. The differences between infection outcomes in 9-10-week-old vs. 7-8-week-old C3H-HeJ mice (with only one 9-10-week-old animal dying while all 7-8-week-old animals died) are striking and underscore the need for a larger sample size. To provide meaningful conclusions, the study requires additional biological replicates (a total of at least two independent experiments) and infection experiments with decreasing doses of leptospires using all three mouse strains. The discussion section could be improved, as many key references are missing. Review articles should be replaced with primary research articles where possible.

**Editorial and Data Presentation Modifications?**

Reviewer #1: Minor

Line 26 – tlr4 is also expressed on other cell types, so I would remove “immune” from this line.

Line 30 – Remove “fully”

Line 80 – After [4] you need a “,”

Line 98 – Change “the LPS receptor” to “which recognizes LPS” as this receptor recognizes other ligands too.

Line 99 – Do you mean “One paper” and not “One mechanism”? Written this is confusing and I think an error.

Line 235 – Add statement about IL1-beta in C57/BL6

Discussion – Reference figures when discussing data (such as line 266-267, 272, etc)

Line 274 – All mice were susceptible to “infection” equally as they were all infected. Change statement to reflect that C3H mice were more susceptible to “disease manifestations” or similar.

Figure 1 legend: “Weightloss” should be “weight loss”

Fix references to Figures in Text: There is no Figure 3C, yet it is referenced in the text.

Reviewer #2: The writing contains awkward sentence structures and grammatical issues that should be addressed for clarity.

**Summary and General Comments:**

Reviewer #1: This paper adds critical information to the field to try and separate the various mouse models of Leptospira infection that are in use. This is of extreme importance to furthering research and comparison between research groups that may use different models of infection. This is novel as no one has done a direct comparison of these mouse strains to date. Therefore, this is critical for the field. For convenience, I am copying all of my major and minor revisions below (which includes the experiments that need to be completed). Overall, I feel this paper should be published after the following revisions are addressed:

1. The authors are making the claim that C3H/HeJ mice are the only strain that sustained lethal infection, however, no comparison was ever made using age-matched mice of the other strains. Therefore this statement cannot be made without proper comparisons. Experiments need to be done with 7-8 week mice of the other strains to allow direct comparison.

a. Line 31-32 – this experiment was done with the 7-8 week mice

b. Line 33-36 – This experiment was done with 9-10 week mice

c. The way lines 31-36 are written the authors state the C3H/HeJ mice had lethal infection, but then had similar survival curves - thus contradicting without stating these are different aged mice.

d. Apply this to other instances in the paper. Line 258-259, 276-277 etc.

2. Mouse experiments were only done with 4 mice and two replicates. A third replicate needs to be done for accurate and valid comparisons to be made, as this is the standard for scientific conclusions to be drawn.

3. Figures: Ideally, all three replicates (when the final one is performed) should be averaged and presented as a combined figure. Otherwise you need to provide a strong reason why you aren’t combining them, and include the data from all three in the supplement. This will allow reproducibility to be seen and verified.

4. Line 214 – an explanation of why younger mice were all of a sudden chosen for analysis is needed, and then as above, experiments need to be completed with the other strains with same ages of mice.

5. Figures: Low quality images, when looking at it there seems to be pixelation causing additional faint lines to show up on graphs. Please provide better quality images.

6. Supplementary Figure 1: This would perhaps be best incorporated into Figure 1 so we can see the differences between the two ages of mice easily. This is an important conclusion you are drawing and it is in the supplement.

7. Conclusions: Generally, conclusions are supported. However, as described above there is no evidence that 7-8 week old mice of the C3H/HeN or C57BL/6 mice would not result in a lethal infection. This needs to be addressed experimentally in order for the conclusion that the C3H/HeJ is the only lethal model of infection (and the caveat added on age of mice).

8. In the discussion, a section needs to be added discussing the importance of the RT-qPCR results. These were only referenced in the results section with no discussion of their importance or differences between strains and what impact that would have on the model. Please add this to the manuscript.

Figure 3B: qPCR of a culture following a harvest does not provide quantitative information on the burden in tissue. Cultures do not grow the same and therefore this is not a quantitative measure and should not be included. You can include a statement that cultures were positive, which is what matters.

Minor

Line 26 – tlr4 is also expressed on other cell types, so I would remove “immune” from this line.

Line 30 – Remove “fully”

Line 80 – After [4] you need a “,”

Line 98 – Change “the LPS receptor” to “which recognizes LPS” as this receptor recognizes other ligands too.

Line 99 – Do you mean “One paper” and not “One mechanism”? Written this is confusing and I think an error.

Line 235 – Add statement about IL1-beta in C57/BL6

Discussion – Reference figures when discussing data (such as line 266-267, 272, etc)

Line 274 – All mice were susceptible to “infection” equally as they were all infected. Change statement to reflect that C3H mice were more susceptible to “disease manifestations” or similar.

Figure 1 legend: “Weightloss” should be “weight loss”

Fix references to Figures in Text: There is no Figure 3C, yet it is referenced in the text.

Reviewer #2: This manuscript presents a study comparing infection outcomes in TLR4-hyporesponsive (C3H-HeJ) and TLR4-competent (C3H-HeN, C57BL/6) mice. The study provides valuable insights into host-pathogen interactions, but several aspects require further attention before publication. The fact that C3H-HeJ mice serve as a good model for both sublethal and lethal leptospirosis has already been established in the field, diminishing the novelty of this study. Importantly, given the apparent low number of animals used, it is not possible to conclusively determine whether one mouse strain is more appropriate than another as a sublethal model of leptospirosis.

PLOS authors have the option to publish the peer review history of their article (what does this mean? ). If published, this will include your full peer review and any attached files.

**Do you want your identity to be public for this peer review?** For information about this choice, including consent withdrawal, please see our Privacy Policy .

Reviewer #1: No

Reviewer #2: No

**Figure resubmission:**

**Reproducibility:**



---

## [Editor Report · Decision Letter 1]

20 May 2025

Dear Professor Gomes-Solecki,

We are pleased to inform you that your manuscript 'TLR4 competence and mouse models of sublethal leptospirosis' has been provisionally accepted for publication in PLOS Neglected Tropical Diseases.

Best regards,

Joseph M. Vinetz

Section Editor

Joseph Vinetz

Section Editor

Shaden Kamhawi

co-Editor-in-Chief

Paul Brindley

co-Editor-in-Chief

---

## [Editor Report · Acceptance letter]

Dear Professor Gomes-Solecki,

We are delighted to inform you that your manuscript, "TLR4 competence and mouse models of sublethal leptospirosis," has been formally accepted for publication in PLOS Neglected Tropical Diseases.

Best regards,

Shaden Kamhawi

co-Editor-in-Chief

Paul Brindley

co-Editor-in-Chief
